# Radiometric Method for Determining Canopy Stomatal Conductance in Controlled Environments

**Oscar Monje [1],*** **and Bruce Bugbee [2]**

[1]  AECOM, Air Revitalization Lab, Mail code: LASSO-008, Kennedy Space Center, Merritt Island, FL 32899, USA

[2]  Plants, Soils and Biometeorology Depatment, Utah State University, Logan, UT 84322, USA; bruce.bugbee@usu.edu

*  Correspondence: oscar.a.monje@nasa.gov; Tel.: +1-321-861-2935

**Abstract:** Canopy stomatal conductance is a key physiological factor controlling transpiration from plant canopies, but it is extremely difficult to determine in field environments. The objective of this study was to develop a radiometric method for calculating canopy stomatal conductance for two plant species—wheat and soybean from direct measurements of bulk surface conductance to water vapor and the canopy aerodynamic conductance in controlled-environment chambers. The chamber provides constant net radiation, temperature, humidity, and ventilation rate to the plant canopy. In this method, stepwise changes in chamber $CO_2$ alter canopy temperature, latent heat, and sensible heat fluxes simultaneously. Sensible heat and the radiometric canopy-to-air temperature difference are computed from direct measurements of net radiation, canopy transpiration, photosynthesis, radiometric temperature, and air temperature. The canopy aerodynamic conductance to the transfer of water vapor is then determined from a plot of sensible heat versus radiometric canopy-to-air temperature difference. Finally, canopy stomatal conductance is calculated from canopy surface and aerodynamic conductances. The canopy aerodynamic conductance was 5.5 mol m$^{-2}$ s$^{-1}$ in wheat and 2.5 mol m$^{-2}$ s$^{-1}$ in soybean canopies. At 400 umol mol$^{-1}$ of $CO_2$ and 86 kPa atmospheric pressure, canopy stomatal conductances were 2.1 mol m$^{-2}$ s$^{-1}$ for wheat and 1.1 mol m$^{-2}$ s$^{-1}$ for soybean, comparable to canopy stomatal conductances reported in field studies. This method measures canopy aerodynamic conductance in controlled-environment chambers where the log-wind profile approximation does not apply and provides an improved technique for measuring canopy-level responses of canopy stomatal conductance and the decoupling coefficient. The method was used to determine the response of canopy stomatal conductance to increased $CO_2$ concentration and to determine the sensitivity of canopy transpiration to changes in canopy stomatal conductance. These responses are useful for improving the prediction of ecosystem-level water fluxes in response to climatic variables.

**Keywords:** canopy stomatal conductance; aerodynamic conductance; elevated $CO_2$; climate change

## 1. Introduction

Understanding boundary layer and land surface feedbacks on canopy transpiration is essential for developing simpler and realistic climate change models and for improving the prediction of ecosystem-level water fluxes in response to climatic variables [1,2]. Canopy stomatal conductance ($G_S$), a key physiological factor controlling transpiration from plant stands, is an important component of land surface feedbacks because it regulates evapotranspiration and surface temperature changes in response to incident radiation, $CO_2$ concentration, and vapor pressure deficit (VPD). This regulatory function is reflected in canopy temperature, which in turn, determines the magnitude and direction

of sensible heat exchange between the vegetation and its environment. At regional scales, stomata exert little control, and daily transpiration of well-watered vegetation is predominantly controlled by radiation and temperature [3–5], in part due to feedbacks that cannot be predicted from single leaf measurements alone [3]. Since canopy-scale transpiration is determined by the ratio between canopy aerodynamic conductance ($g_A$) and $G_S$ [4,6,7], improved methods for measuring $g_A$, as well as measuring responses of $G_S$ to environmental variables (e.g., light, $CO_2$, VPD, soil moisture, and temperature), are needed for studying the processes controlling feedback and stomatal control of evaporation from regional land surfaces.

In the field, $g_A$ is often approximated by the conductance to momentum transfer determined using the log-wind profile approximation, which requires at least 100 m of fetch and thus cannot be used in controlled-environment chambers [8]. In controlled environments, leaf boundary conductance has been estimated from measurements with wet filter paper analogs [9], from cooling curves of metal models of leaves [10], or from combined energy balance and temperature measurements using metal leaf models [11,12]. However, Jarvis and McNaughton [3] argue that leaf level measurements of stomatal control of transpiration may not be applicable to plant canopies in the field because the amount of ventilation in leaf cuvettes and plant chambers typically prevents feedback between transpiration and VPD observed in the field.

$G_S$ can be derived using energy balance approaches from canopy surface conductance to water vapor ($G_{SFC}$), latent heat flux (LE), and the VPD at the leaf surface (Ds). Similarly, the single-layer or "big leaf" $G_S$ may be computed from $G_{SFC}$, provided the boundary layer conductance to water vapor (i.e., $g_A$) and the mean aerodynamic canopy temperature ($T_{Aero}$) are known [13,14]. In the field, $G_{SFC}$ is calculated from canopy-level LE obtained using lysimeters, Bowen ratio, and eddy correlation systems [15], or by inverting the Penman–Monteith equation [13]. However, these approaches for measuring canopy-level LE do not permit partitioning of transpiration among individual species and often cannot distinguish between transpiration and evaporation from the soil or from wet leaf surfaces. Thus, $G_{SFC}$ is not always related to estimates of canopy $G_S$ derived from single leaf measurements because it often includes significant contributions from soil evaporation [13].

Smith et al. [16] used canopy-level energy balance measurements to estimate sensible heat flux (H) of a wheat field from radiometric canopy temperature when canopy $g_A$ and LE were known. Their approach produced accurate estimates of hourly LE, which suggests that $g_A$ could be estimated if H and the canopy-to-air temperature difference are measured accurately. However, canopy $g_A$ determined from changes in radiometric canopy temperature differs from $g_A$ determined using the log-wind profile approximation because it includes the conductance to heat and water vapor across leaf boundary layers, as well as the turbulent conductance caused by the movement of air eddies between the canopy and the atmosphere [17,18].

Canopy $G_S$ obtained from energy balance approaches may contain considerable errors because Ds and LE are estimated using measurements of canopy radiometric temperature ($T_R$) to approximate the aerodynamic canopy temperature [19–21]. In the field, estimating $T_{Aero}$ from infrared measurements is complicated because radiometric measurements depend on the view angle of the sensor, sun angle, degree of crop cover, spatial variability of canopy emissivity, and atmospheric attenuation, and they often include significant temperature contributions from soil surfaces [20,22–25]. A systematic difference of $-1\,°C$ was measured between radiometric and aerodynamic temperatures by Huband and Monteith [26], although differences ranging from 2 to 6 °C have also been observed [13]. The difference between $T_{Aero}$ and $T_R$ can be very small in dense canopies, but it can exceed 10 °C in sparse vegetation because of contributions from soil temperature ([18,27]. These differences are significant because small errors of $-1\,°C$ in the surface-to-air temperature difference can represent an uncertainty in latent heat fluxes of ~40 W m$^{-2}$ [13]. Many complicating factors that affect infrared measurements of canopy temperature in field settings can be minimized in controlled environments by using high planting density canopies grown under constant lighting. In dense canopies, canopy brightness temperature measured with infrared sensors approximates canopy radiometric temperature [28], but errors due

to radiation reflected into the sensor and artifacts caused by fluctuating sensor body temperatures remain [29].

The purpose of this study was to develop a radiometric method for measuring canopy $G_S$ of well-watered plant canopies in controlled environments. The hypothesis tested was that a radiometric method utilizing canopy-level energy balance measurements provides more accurate estimates of canopy stomatal conductance than bottom-up methods scaling leaf-level to canopy-level conductance or top-down methods that estimate canopy surface conductance from field data. Bottom-up methods require that leaf area index be known and must integrate the responses of leaf stomatal conductance to vertical gradients in radiation, temperature, and humidity. Conductances from top-down methods using field data typically include significant contributions of soil evaporation, and field radiometric data include soil surface temperatures that cause significant differences between radiometric and aerodynamic temperatures [13].

Simultaneous measurements of energy balance, gas fluxes, and canopy temperature at constant environmental conditions were used to compute canopy $G_S$ from surface $G_{SFC}$ and canopy $g_A$ (Figure 1). The relation between radiometric and aerodynamic temperatures was studied by varying incident radiation and wind speed. Canopy $G_S$ and $g_A$ of high planting density wheat (*Triticum aestivum* L. cv. USU Apogee) and soybean (*Glycine max* L. cv. Hoyt) canopies were measured at 400 umol mol$^{-1}$ $CO_2$. The radiometric method was used to explore the effects of rising $CO_2$ concentration on canopy $G_S$ and to describe stomatal feedbacks to transpiration using the canopy-scale decoupling coefficient.

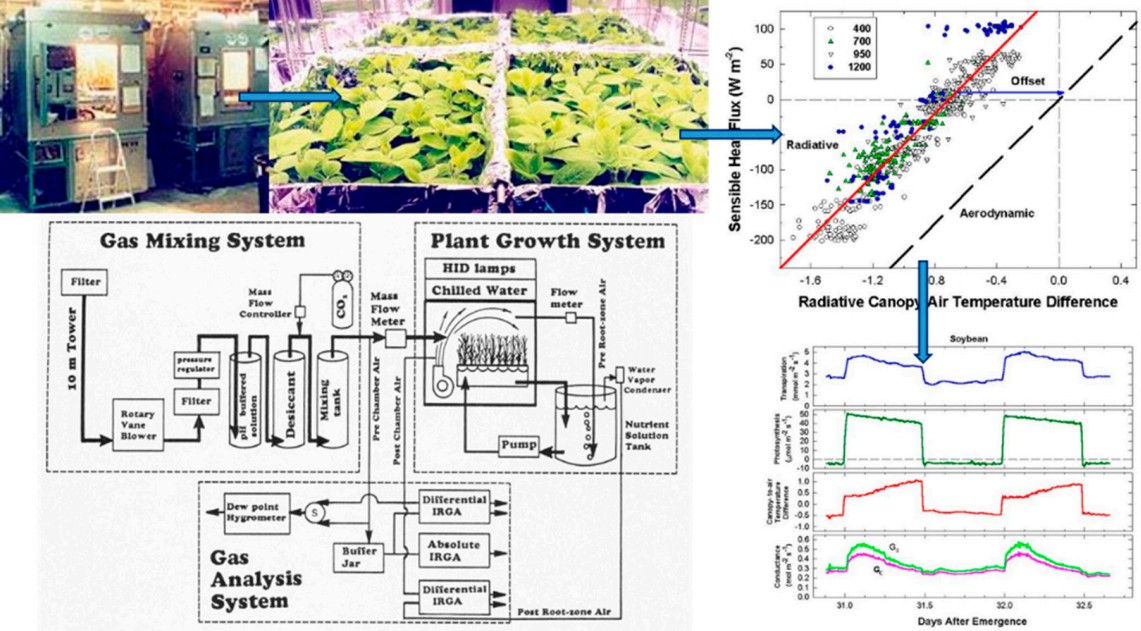

**Figure 1.** A two chamber, open gas exchange system capable of determining canopy aerodynamic conductance from measures of sensible heat flux and canopy-to-air temperature difference was used to calculate canopy stomatal conductances of wheat and soybean canopies.

## 2. Materials and Methods

### 2.1. Chamber System

In this study, 18–35-day-old, closed wheat and soybean canopies were used to examine various aspects of the method—energy balance responses to changes in radiation forcing, responses of vertical gradients in canopy-to-air temperature to fan speed or light level, responses in canopy-to-air temperature and sensible heat flux to $CO_2$ concentration, etc. Each test took several days to conduct, and plant canopies of different ages were used because the logistics of growing canopies to the same age for each test was impractical. Thus, conductances observed in a vegetative 20-day-old wheat

canopy may not be the same as in a reproductive 35-day-old canopy due to ontogenetic changes in canopy structure (i.e., the presence of heads). However, overall the method is robust as long as energy balance components and canopy-to-air temperature differences are measured accurately and simultaneously.

### 2.2. Cultural and Environmental Conditions

Wheat and soybean canopies were grown in sealed, water-cooled, controlled-environment chambers (Model EGC-13, Environmental Growth Chambers, Chagrin Falls, OH, USA). Two canopies of the same species were grown simultaneously in adjacent chambers. Wheat was seeded into lids containing a 10 mm layer of inert media (Isolite, size CG-2, Sumitomo Corp., Denver, CO, USA) at a density of 1100 plants $m^{-2}$. Soybean seedlings were transplanted into closed cell foam plugs in a Styrofoam lid at a planting density of 60 plants $m^{-2}$. The seedling roots grew into a recirculating hydroponic solution after germination. The hydroponic system is described in Monje and Bugbee [30].

Inside each chamber, a polished aluminum, reflective side-wall was built around the perimeter of the ~1 $m^2$ canopy to minimize edge effects and side lighting. The incident photosynthetic photon flux ($PPF_o$) was 1600 $\mu$mol $m^{-2}$ $s^{-1}$ for wheat and 750 $\mu$mol $m^{-2}$ $s^{-1}$ for soybean. Lighting was provided by four, 1000 W high-pressure sodium (HPS) lamps, which were adjusted with neutral density filters to achieve $\pm$5% $PPF_o$ uniformity over the crop surface. $PPF_o$ was measured at the top of the canopy with a quantum sensor (Model LI-190SB, LI-COR, Lincoln, NE, USA), and was adjusted daily throughout the life cycle by lowering the canopy platform as the plants grew taller. Longwave radiation emitted by the lamps was removed by a 10 cm deep water filter. The filter consisted of a glass box filled with recirculating, chilled water located below the lamps. The water filter under the lamps was removed over the course of several days during tests that change surface radiation forcing by increasing incident $PPF_o$ and longwave radiation impinging on the canopy. Advective conditions existed in the chamber because the temperature control system heated the air to maintain the chamber temperature setpoint, and the canopy was exposed to a continuous flow of warm air.

Air temperature was 21.0 $\pm$ 0.3 $^\circ$C, the barometric pressure was 86 $\pm$ 0.1 kPa, and chamber $CO_2$ varied between 400 and 1400 $\mu$mol $mol^{-1}$ to manipulate canopy temperature, LE and H. Relative humidity at night was 50% $\pm$ 5%. During the day, transpiration humidified the 1300 L of chamber air and daytime relative humidity was 70 $\pm$ 5%. Wheat was grown under a 20 h light/4 h dark photoperiod and soybean under a 12 h light/12 h dark photoperiod. The canopies grew in a chamber supplied with a constant temperature ($T_{Air}$) and VPD of bulk air surrounding the vegetation ($D_{Bulk}$) as well as a constant wind speed. Thus, boundary layer forcing ($T_{Air}$ and $D_{Bulk}$) and surface layer feedbacks (chamber wind speed) were held constant, but in nature they are dictated by diurnal changes in local climate ($T_{Air}$, $D_{Bulk}$, and wind speed).

### 2.3. Gas Exchange System

Each chamber used an open gas exchange system to measure canopy photosynthesis [30,31]. The open flow system ensured that humid air (~50% relative humidity) of a constant $CO_2$ concentration (setpoint $\pm$10 $\mu$mol $mol^{-1}$) fed the chambers at flow rates between 500 and 1100 L $min^{-1}$. Air mass flow (MF; mol $s^{-1}$) into the chambers was measured with mass flow meters (Model 730, Sierra Instruments, Monterey, CA, USA). The gas exchange systems were modified to use a dew point hygrometer to measure the water vapor concentration of pre- and post-chamber air from which evapotranspiration was calculated. Two solid-state multiplexers (Model AM-25T, Campbell Scientific, Logan, UT, USA), each referenced to a 100 Ohm platinum resistance thermometer, were used for precision thermocouple measurements. Data acquisition and control were performed with a datalogger (Model CR-10T, Campbell Scientific, Logan, UT, USA).

Gas exchange fluxes in each chamber were measured continuously and averaged for 2 min every 8 min. Net photosynthesis, $P_{net}$, and dark respiration rates were calculated from the difference between pre- and post-chamber $CO_2$ concentrations ($\Delta CO_2$), multiplied by MF of air into the chambers. $\Delta CO_2$

was measured with a differential infrared gas analyzer (Model LI-6251, LI-COR, Lincoln, NE, USA). The temperature, water vapor band broadening, and dilution corrections used for the C fluxes are described in Monje and Bugbee [30]. Chamber evapotranspiration (ET) was determined from the difference in mole fraction of water vapor between pre- and post-chamber air ($\Delta X_{h20}$), multiplied by mass flow rate entering the chamber (ET = $\Delta X_{h20} \times$ MF). $\Delta X_{h20}$ was determined from sequential measurements of pre- and post-chamber air dewpoint made with a dewpoint hygrometer (Model Dew-10, General Eastern, Watertown, MA, USA). Air flow was changed to increase $\Delta CO_2$ and $\Delta X_{h20}$ in the chamber. The flow rate of air entering the chamber was not corrected for the amount of water vapor added by canopy transpiration (a maximum of ~15 L/day) because this correction was negligible, which would not be the case in smaller leaf gas exchange systems [32]. Water use efficiency ($\mu$mol mmol$^{-1}$) was calculated from the ratio of $P_{net}$ to ET.

*2.4. Chamber Wind Speed*

Wind speed above and within the canopies was measured with heat transfer needle anemometers (Model AN-27, Soiltronics, Burlington, WA, USA). These anemometers were well-suited for making wind measurements within canopies because they are small, have fast response times (half-life $t_{1/2} = 1$ s), and are omnidirectional. The anemometers were calibrated in a wind tunnel for windspeeds between 0.05 and 5 m s$^{-1}$ [33]. Vertical gradients in mean wind speed above and within the canopies were measured with anemometers spaced between 4 and 6 cm apart. Each chamber was modified to include variable speed centrifugal blowers so that wind speed above the vegetation could be controlled over a wide range. Three wind speed settings (high: 2.3 m s$^{-1}$; medium: 1.7 m s$^{-1}$, and low: 0.8 m s$^{-1}$) were used in the chambers, but the majority of the measurements were made at the medium setting.

*2.5. Temperature Measurements*

The temperature sensor used to control chamber air temperature was situated 20 cm above the canopy and 10 cm below the lamps. This reference location was chosen because the lamps were found to heat the air in the top 5 cm of the chamber near the water filter. Mean air temperature ($T_{air}$) at the reference location was measured using a shielded and aspirated thermocouple (Type-E, 30 gauge). Vertical profiles of air temperature within the canopies were measured with an aspirated thermocouple manifold. The thermocouples were arranged in parallel within a manifold that held the thermocouples evenly spaced (10 cm apart) and were ventilated at about 1–2 m s$^{-1}$ by a single aspirator (a vacuum cleaner). The aspirated thermocouples were shielded from incident radiation by plastic tubing wrapped in aluminum foil. The vertical profiles in temperature were expressed as an air temperature difference from the reference air temperature above the canopy.

Canopy temperature measurements made using infrared temperature sensors are described using the nomenclature and definitions of Norman and Becker [28]. Two nadir-viewing (e.g., perpendicular to the canopy) infrared sensors in each chamber (Model IRTS-P, Apogee Instruments, Logan, UT, USA) were used to measure canopy brightness temperature ($T_{canopy,IR}$), which is a directional temperature that depends on the angle of observation, the wavelength band of the infrared sensor, the sensor body temperature, and sensor position above the top of the canopy. The IRTS-P infrared sensors have a 90° field of view and an accuracy of ±0.2 °C. The 8–14 $\mu$m wavelength band was viewed. They were placed in the center of the canopy at a height of 10 cm above the foliage, where the chamber walls could not be seen. The calibration procedures, the field of view considerations, and the functions used to correct for sensor body temperature for these sensors are described in Bugbee et al. [29].

Canopy $T_{Aero}$, formally defined as the extrapolation of air temperature profile down to an effective height within the canopy at which the vegetation components of sensible and latent heat flux arise [18], is the mean canopy temperature felt by the air that solves the energy balance equation exactly. $T_{Aero}$ cannot be measured directly. It can be obtained from H when $T_{air}$ and $g_A$ are known, but is typically approximated by $T_R$, the canopy radiometric temperature [13,27]. Canopy $T_R$ was derived from $T_{canopy,IR}$ after correcting for the sky irradiance (e.g., proportional to sky temperature, $T_{Sky}$) that is

reflected by the canopy and by the chamber walls into the field of view of the infrared sensor [28]. In the controlled-environment chambers, sky irradiance is emitted by the warm chamber surface areas above the canopy, which were proportionally divided into a 20% chamber wall and an 80% glass water filter. The difference between $T_R$ and $T_{canopy,IR}$ depends on the canopy emissivity, $\varepsilon_c$ (Equation (1)):

$$\sigma\, T^4_{canopy,IR} = \varepsilon_c{}^*\sigma^*T^4_R + (1 - \varepsilon_c)^*\sigma^*T^4_{Sky} \tag{1}$$

where $\sigma$ = Stefan–Boltzman constant (W m$^{-2}$ K$^{-4}$), and $T_{Sky}$ = temperature of the chamber surfaces above the canopy (K). In this paper, it was assumed that $T_R \approx T_{canopy,IR}$ because the correction for canopy emissivity is small ($\approx$ 0.2 °C). For example, if $T_{glass}$ = 30 °C, $T_{wall}$ = $T_{air}$ = 23 °C, $T_{canopy,IR}$ = 24 °C (e.g., 20% wall and 80% glass temperature), and $\varepsilon_c$ = 0.97, then the difference between $T_R$ and $T_{canopy,IR}$ is only 0.14 °C. If the water filter under the lamps is removed, $T_R$ increases by ~0.5 °C, $T_{glass}$ = 45 °C, and the difference between $T_R$ and $T_{canopy,IR}$ rises to ~0.5 °C (Equation (1)). These conditions are unique to controlled-environment conditions because such high $T_{Sky}$ temperatures are never observed in the field.

*2.6. Absorbed Radiation*

Energy exchange and photosynthesis are proportional to the amount of radiation absorbed by plant canopies, which is determined by the direct beam fraction of incident radiation, the canopy structure, and the optical properties of the plant elements [34]. Incident $PPF_o$ and shortwave radiation within the growth chamber were measured at canopy height. Shortwave radiation between 0.285 and 2.8 μm was measured with a precision spectral pyranometer (The Eppley Laboratory, Model PSP, Newport, RI, USA). Incident non-photosynthetic, shortwave radiation ($NPSW_o$) was determined by subtracting $PPF_o$ (converted to energy units assuming 5 μmol m$^{-2}$ s$^{-1}$ per W m$^{-2}$ for HPS lamps) from the total shortwave radiation. The fraction of PPF absorbed by the canopy ($PPF_{abs}$) was calculated from the product of radiation capture and $PPF_o$, as described by Monje and Bugbee [30]. A diffuse light fraction of 0.7 was measured in the chamber using a shadow band to shield the quantum sensor from direct radiation. The fraction of non-photosynthetic, shortwave radiation absorbed by the canopy ($NPSW_{abs} = (1 - \varrho_c) NPSW_o$) depends on the canopy reflection coefficient (or surface albedo), $\varrho_c$, in the near-infrared (NIR). $\varrho_c$ was estimated using Equation (2) from the single leaf scattering coefficient ($\sigma_S$) [35]:

$$\varrho_c = [1 - (1 - \sigma_S)^{1/2}]/[1 + (1 - \sigma_S)^{1/2}]. \tag{2}$$

$\sigma_S$ varies with the wavelength of the radiation and equals the sum of the fractions of reflected and transmitted light. In the visible spectrum, the $\varrho_c$ of the high planting density wheat canopies was 0.055 during vegetative growth [36], which corresponds to a $\sigma_S$ of 0.2. Since the NIR $\varrho_c$ was not measured directly, it was derived from Equation (2) assuming an NIR $\sigma_S$ of 0.8. For comparison, the single leaf reflectance (0.43) and transmittance of winter wheat (0.33) in the NIR combine to give an NIR $\sigma_S$ of 0.76 [37]. Thus, $NPSW_{abs}$ was 0.62 × $NPSW_o$ for a $\varrho_c$ of 0.38 in the NIR. Although this approximation overestimates $\varrho_c$ in sunny conditions (e.g., high direct beam radiation), it predicts it accurately under overcast conditions [38], similar to the highly diffuse radiation found in these controlled-environment chambers.

*2.7. Net Radiation, Evapotranspiration, and Photosynthesis*

The net radiation above the canopy, $R_{net}$, was assumed proportional to net input of shortwave radiation and incoming longwave radiation (Equation (3)).

$$R_{net} = PPF_{abs} + NPSW_{abs} + {\downarrow}L_g - {\uparrow}L_c \tag{3}$$

where $PPF_{abs}$ = absorbed photosynthetic radiation (W m$^{-2}$), $NPSW_{abs}$ = absorbed non-photosynthetic shortwave radiation (W m$^{-2}$), ${\downarrow}L_g$ = longwave radiation emitted by the glass from the water filter and

the chamber walls (W m$^{-2}$), and $\uparrow L_c$ = longwave radiation emitted by the canopy (W m$^{-2}$). Assuming that the longwave radiation components ($\downarrow L_g - \uparrow L_c = \varepsilon_c \downarrow L_g - \varepsilon_c \sigma T^4_R = \varepsilon_c \sigma T^4_{Sky} - \varepsilon_c \sigma T^4_R$) nearly canceled each other was acceptable as long as the differences between $T_{Sky}$ and $T_R$ were also small. For example, if $T_{glass}$ = 30 °C, $T_{wall}$ = $T_{air}$ = 23 °C, and $T_R$ = 24 °C, then $T_{sky}$ = 28.6 °C, and $\downarrow L_g - \uparrow L_c$ = 27 W m$^{-2}$. Although Equation (3) ignores changes in longwave radiation within the canopy caused by vertical gradients in temperature, it was a better estimate than direct measurements with a net radiometer. Most net radiometers are calibrated for field operation, where the fraction of longwave radiation is much smaller than in these chambers, and the dimensions of the chambers placed the net radiometer close to the top of the foliage, where self-shading led to significant overestimates of the net radiation flux. Net radiometers are preferred in chambers illuminated by solar radiation, but they are affected during cloudy days with highly diffuse radiation.

Net radiation in the chamber could be varied by either changing PPF$_o$ with neutral density filters (window screen filters) or by draining the water filter under the lamps. Shading with neutral density filters does not alter the spectral composition of the incident radiation. In contrast, the water filter under the lamps reduces the amount of longwave radiation impinging on the canopy, thereby increasing the ratio of PPF$_{abs}$ to R$_{net}$ [39]. Removing the water filter increased $\downarrow L_g$ compared to $\uparrow L_c$ and added ~100 W m$^{-2}$ to R$_{net}$, as the glass temperature measured with a thermocouple reached 45 °C. The PAR$_{abs}$ to R$_{net}$ ratio was 83% of R$_{net}$ in a chamber with a water filter below the lamps, but was only 64% of R$_{net}$ when the water filter was removed. These changes in surface radiation forcing (R$_{net}$) were used to change canopy temperature and H for studying the relation between T$_{Aero}$ and T$_R$.

Chamber ET (mmol m$^{-2}$ s$^{-1}$) consisted of canopy transpiration (E$_{can}$) and evaporation (E) from the hydroponic solution through the porous media sustaining the plants (Equation (4)).

$$ET = E_{can} + E. \tag{4}$$

Chamber latent heat flux (LE; W m$^{-2}$) was determined from the product of ET and the heat of vaporization of water (44 kJ mol$^{-1}$). Evaporation from the hydroponic tubs, covered with lids but without a canopy, was small (~2% of R$_{net}$ when expressed in W m$^{-2}$). This made ET essentially equal to E$_{can}$ in this study and ensured that T$_{Aero}$, calculated from the energy balance measurements, was mostly due to the flux of sensible heat between the foliage and the air flowing above the canopy.

In controlled environments, P should be included in the energy balance equation at high light intensities because it becomes a large fraction of R$_{net}$. Photosynthesis (P; W m$^{-2}$), the conversion of energy in radiation into stored chemical energy, was derived from the product of canopy photosynthesis, P$_{net}$ [30], and the enthalpy of combustion for CHO (479 KJ mol$^{-1}$) [40].

*2.8. Canopy Sensible Heat Flux*

In the steady state, H is the energy exchanged by conduction and convection between the canopy and the chamber air. The canopy energy balance equation was rearranged for calculating H by residual (Equation (5)), where R$_{net}$ = net radiation, LE = latent heat flux, G = soil heat flux, and P = energy storage in photosynthesis.

$$H = R_{net} - LE - G - P. \tag{5}$$

LE includes water vapor fluxes mostly due to canopy E$_{can}$ because evaporation was only 2% of R$_{net}$. The soil heat flux, G, is a component of land surface feedbacks that depends on the amount of energy available below the canopy. G was assumed to be zero due to a poor transfer of heat through the dense canopies (high planting densities and leaf area indices > 15) used in this study, but this may not be a valid assumption during early development when the plants are seedlings. P was determined from canopy photosynthesis, which can be as much as 10% of R$_{net}$ at high light intensities. For example, if P$_{net}$ = 60 umol m$^{-2}$ s$^{-1}$ at a PPF$_o$ of 1400 µmol m$^{-2}$ s$^{-1}$, then P = 29 W m$^{-2}$. Equation (5) allows for a comparison of energy fluxes in common energy units (W m$^{-2}$) and allows H to be determined by residuals. However, Equation (5) ignores the thermal storage within the canopy, which is small for

the short vegetation used in this study, but this storage can be as high as 5–10% of the net radiation in forest canopies [13].

### 2.9. Canopy Aerodynamic Conductance

In field settings, the log-wind profile approximation allows canopy $g_A$ to be determined from H provided $\Delta T_A$, the aerodynamic canopy-to-air temperature difference ($\Delta T_A = T_{Aero} - T_{air}$), the displacement height, and the roughness length are known [41]. However, the short fetch (1 m) of the canopies used in this study precludes the use of the log-wind profile approximation for calculating $g_A$ in controlled-environment chambers. Instead, an analog of Ohm's law (Equation (6)) that relates the surface-to-air temperature difference to the sensible heat loss from the surface was used to describe the energy transfer between the canopy and the chamber air [8]:

$$H = \varrho {}^{*}Cp{}^{*}g_A{}^{*}(T_R - T_{air}) \tag{6}$$

where $\varrho$ = density of air (kg m$^{-3}$), Cp = heat capacity of air at constant pressure (kJ m$^{-3}$ °C$^{-1}$), $g_A$ = canopy aerodynamic conductance (mol m$^{-2}$ s$^{-1}$), and $T_R$ (°C) was approximated by $T_{canopy,IR}$. $T_{air}$ was measured at the reference height above the canopy, and used to determine the radiometric canopy-to-air temperature difference ($\Delta T_{IR} = T_R - T_{air}$). Equation (6) assumes that the slope between H and $\Delta T_{IR}$ equals the slope between H and $\Delta T_A$, when $T_{Aero} = T_R$. This assumption is valid for fully covered canopies, whereby the contribution to $\Delta T_{IR}$ from the temperature of the surface below the vegetation (e.g., soil or hydroponic tray) is negligible.

The canopy leaf boundary layer conductance component depends on leaf shape and size, and the turbulent conductance component depends on wind speed and canopy aerodynamic roughness [8]. Canopy aerodynamic conductances of dense wheat and soybean canopies with distinct canopy architectures were calculated from the slopes of plots of H vs. measured $\Delta T_{IR}$ following Equation (6). Radiometric $\Delta T_{IR}$ and H were varied simultaneously by manipulating chamber $CO_2$ concentration at constant environmental conditions (wind speed and VPD) over the course of several days. The $g_A$ measured for each species results from the amount of drag generated by the interaction between canopy architecture and the chamber air recirculating at constant wind speed.

Although changes in $CO_2$ affect H and $\Delta T_{IR}$ through changes in stomatal conductance, $g_A$ remains constant at a fixed chamber wind speed. The highly turbulent conditions in the chamber ensure that free convection effects are negligible compared to forced convection, so changes in light level should not significantly affect canopy $g_A$. Estimates of $g_A$ obtained from the slope of a plot of H vs. $\Delta T_{IR}$ are also insensitive to systematic errors in H (e.g., offset errors in $R_{net}$) because these do not affect the slope. In this context, the canopy $g_A$ obtained by this radiometric method represents the canopy leaf boundary layer conductance, as well as the conductance for turbulent heat transfer between the leaves at $T_{Aero}$ and $T_{air}$ measured at the reference height above the canopy.

The H vs. $\Delta T_{IR}$ plot is also useful for exploring differences between $T_{Aero}$ and $T_R$. The offset, defined as the value of $\Delta T_{IR}$ when H and $\Delta T_A$ are zero (Equation (7)), quantifies this difference because $\Delta T_{IR}$ and $\Delta T_A$ are referenced to a common $T_{air}$.

$$\Delta T_A = \Delta T_{IR} + \text{Offset}. \tag{7}$$

The behavior of Offset was studied by varying the intensity of the radiation incident on the canopy using neutral density filters and by changing the chamber wind speed. These changes effectively alter surface radiation forcing (PPF$_o$) and surface layer feedbacks (wind speed).

### 2.10. Canopy $G_{SFC}$ and $G_S$

The measurement of canopy ET in controlled environments makes it possible for calculating a "big-leaf" surface canopy conductance ($G_{SFC}$) with a corresponding effective VPD at the "big-leaf" surface ($D_S$). Surface $G_{SFC}$ was calculated from the ratio of $E_{can}$ to $D_S$ (Equation (8)):

$$G_{SFC} = E_{can} \times P_{atm}/D_S \qquad (8)$$

where $G_{SFC}$ = canopy surface conductance, $E_{can}$ = canopy transpiration measured using the gas exchange system (mmol m$^{-2}$ s$^{-1}$), and $P_{Atm}$ = atmospheric pressure. $D_{Bulk}$ was calculated using $T_{Air}$ measured at the reference location above the canopy. $D_{Aero}$ is the VPD of the air within the canopy at $T_{Aero}$. When $D_S = D_{Aero}$ in Equation (8), each leaf surface is at the mean aerodynamic temperature and sees the same saturation deficit at its surface, which treats the canopy as a giant single leaf where the average canopy leaf temperature equals $T_{Aero}$.

Canopy $G_{SFC}$ calculated from Equation (8) includes canopy $G_S$ and $g_A$ [10] because these conductances are additive in series. Canopy $G_S$ was calculated from surface $G_{SFC}$ and $g_A$ using Equation (9), the resistance subtraction method [7]. $G_S$ is metabolically controlled canopy stomatal conductance that influences land atmosphere interactions via land surface feedbacks.

$$Gs = G_{SFC} \times g_A/(g_A - G_{SFC}) = ((1/G_{SFC}) - (1/g_A))^{-1}. \qquad (9)$$

*2.11. Canopy Decoupling Coefficient*

At the canopy level, relative magnitudes of $G_S$ and $g_A$ determine the effect of changes in stomatal conductance on the transport of heat and water vapor from an average leaf surface, through leaf and canopy boundary layers to an effective sink for heat and water vapor above the canopy [3]. The boundary layer surrounding vegetation allows transpired water vapor to humidify air near the leaf surface (e.g., it lowers $D_S$ compared to $D_{Bulk}$), altering the driving force for transpiration; thus, $E_{can}$ becomes less sensitive to changes in stomatal conductance. This feedback between $E_{can}$ and $D_S$ is important for diminishing the sensitivity of $E_{can}$ to proportional changes in $G_S$ [1,3,4].

The dimensionless decoupling coefficient, $\Omega$, quantifies the sensitivity of $E_{can}$ to changes in stomatal aperture and depends on the influence that $G_S$ and $g_A$ exert on how closely conditions at the leaf surface (e.g., $D_S$) are linked to $D_{Bulk}$ of the free air stream. Equation (10) calculates $\Omega$ from $g_A$, $G_S$, and $\varepsilon = s/\gamma$, where s = the slope of the saturation vapor pressure versus temperature, and $\gamma$ = the psychrometric constant [10].

$$\Omega = (\varepsilon + 1)/[\varepsilon + 1 + (g_A/G_S)]. \qquad (10)$$

Equation (10) assumes that the available energy is independent of surface temperature and neglects changes in leaf temperature due to changes in stomatal conductance [4]. In spite of this simplification, $\Omega$ is useful for (1) exploring how differences in canopy architecture (e.g., wheat and soybean) affect canopy transpiration and (2) quantifying the sensitivity of $E_{can}$ to changes in stomatal conductance. Typical values for $g_A$, $G_S$, and $\Omega$ for crops and forests are depicted in Table 1. The magnitude of $\Omega$ effectively determines whether Ecan is primarily controlled by stomata or by the supply of energy. Generally, forests are well coupled, and their transpiration rate is accurately predicted by the Priestley–Taylor equation [3,4]. The sensitivity of transpiration to stomatal control, $dE_{can}$, is determined by the degree of coupling $(1 - \Omega)$ between $D_S$ and $D_{Bulk}$ (Equation (11); [3,6,7]).

$$dE_{can} = (1 - \Omega) \times (E_{can}/G_S) \times dG_S. \qquad (11)$$

**Table 1.** Typical land surface properties that influence the control of transpiration rate from conifers or crops.

| Species | Coupling | $T_{Aero} - T_{air}$ | $g_A$ | $\Omega$ | Relative Magnitude | Transpiration Control |
|---------|----------|-----------|-------|----------|--------------------|-----------------------|
| Conifer | coupled | small | low | ~ 0.1 | $g_A \gg G_S$ | Radiation $\approx \Delta R_{net}$ |
| Crop | decoupled | large | high | ~ 0.8 | $g_A \ll G_S$ | Stomatal $\approx \Delta G_S$ |

*2.12. Responses of Transpiration to Elevated CO$_2$*

Responses of transpiration to $CO_2$ concentration were measured at a constant $PPF_o$ using the same vegetative wheat canopy over a span of 8 days. During this time, chamber $CO_2$ was increased in

a stepwise fashion from 400, to 700, to 950, and to 1200 umol mol$^{-1}$. Canopy gas exchange fluxes and energy balance components were held at each $CO_2$ concentration for 48 h to allow the incremental buildup of sugar pools in tissues throughout the canopy. These data were used to measure canopy aerodynamic conductance and to determine the response of canopy transpiration to increased $CO_2$ concentration. Daily average values of $E_{can}$, $P_{net}$, LE, H, $G_S$, $\Omega$, and WUE were calculated because $G_S$ and $E_{can}$ did not remain constant throughout the day due to diurnal changes in stomatal conductance.

## 3. Results

### 3.1. Wind and Temperature Profiles

Average wind speed and air temperature profiles were measured at different heights above and within wheat and soybean canopies in a ventilated chamber. The mean wind speed at any given plane above the canopy was highly spatially and temporally variable, typically ranging from 0.5 to 2.4 m s$^{-1}$ in wheat (Figure 2A), and from 0.4 to 1.4 m s$^{-1}$ in soybean (Figure 2B). The average wind speed at the canopy surface was attenuated rapidly within the first few centimeters of foliage. Wind speed within the canopies was more uniform than above and was often below 0.4 m s$^{-1}$, reaching as low as 0.1 m s$^{-1}$ at the bottom of the wheat canopy.

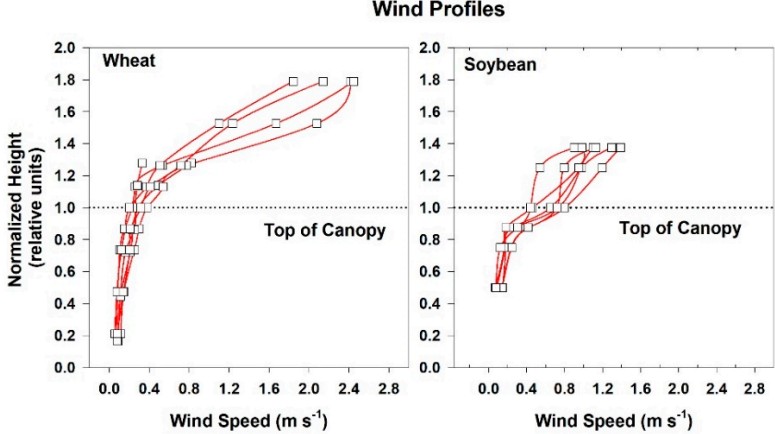

**Figure 2.** Vertical wind profiles of wheat and soybean canopies measured with needle anemometers. Y-axis units are fraction of canopy height.

Vertical air temperature profiles within the growth chamber were homogeneous in an empty, dark chamber since there was no foliage to trap pockets of air, and because the surfaces within the chamber (glass and chamber walls, and the surface of the growth media) equilibrated at nearly the same temperature. When the lights were turned on, the vertical air temperature profiles within the canopy were spatially variable; air near the plants could be 1–5 °C higher than the reference air temperature measured above them depending on the relative magnitudes of incident radiation or wind speed within the chamber (Figure 3). These large air temperature differences within the canopies result from vertical differences in light intensity, leaf temperature, and leaf transpiration rates. Transpiration cools cooled the lower layers of the canopy to temperatures below the reference air temperature, and the uppermost leaf layers remain warmer because they are heated by the absorption of incident radiation.

Incident PPF affected the air temperature difference above and within a wheat canopy (Figure 3A; 20-day old; [$CO_2$] = 400 μmol mol$^{-1}$; $T_{air}$ = 21 °C; RH = 68%; medium wind speed: 1.7 m s$^{-1}$). In the dark, the top layers of foliage remained warmer than the lower layers because they were heated by sensible heat flux from the warm chamber air flowing above the canopy. During the photoperiod, the top of the canopy remained hotter than the lower leaf layers as the top layers of foliage absorbed most of the incident radiation. The air within the top 5 cm of the canopy became hotter than the reference air temperature as incident light levels increased to 1050 and 1850 μmol m$^{-2}$ s$^{-1}$ (Figure 3A).

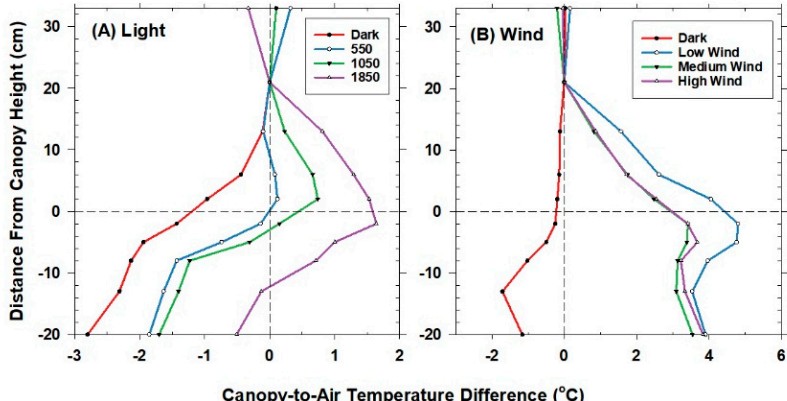

**Figure 3.** Vertical canopy-to-air temperature difference profiles of wheat canopies were affected by (**A**) light intensity and (**B**) chamber wind speed.

At constant PPFo, the fan speed setting (high: 2.3 m s$^{-1}$; medium: 1.7 m s$^{-1}$; low: 0.8 m s$^{-1}$) changed the amount of forced convection in the chamber and affected the vertical air temperature profiles above and within the wheat canopy (Figure 3B; 35-day old; PPF = 1800 μmol m$^{-2}$ s$^{-1}$; $[CO_2]$ = 1200 μmol mol$^{-1}$; $T_{air}$ = 21 °C; RH = 68%). At the low chamber wind speed setting (0.8 m s$^{-1}$), the upper 7 cm of the canopy was 1.5 °C warmer than at medium (1.7 m s$^{-1}$) and high (2.3 m s$^{-1}$) settings (Figure 3B). Air temperature up to 12 cm above the canopy was also heated by 0.5–1.2 °C by the warm foliage at the low wind speed. This suggests a threshold in turbulence in the chamber, above which an increase in wind speed does not continue to affect canopy-air heat exchange.

### 3.2. Diurnal Changes in Energy Balance Components

Sensible heat flux (Figure 4; pink line) was calculated from direct measurements of canopy energy balance components (net radiation—red line; latent heat—blue line; photosynthesis—green line) in wheat (18-day-old; $[CO_2]$ = 680 μmol mol$^{-1}$; PPF = 1600 μmol m$^{-2}$ s$^{-1}$; $T_{air}$ = 21 °C; RH = 68%) and soybean (25-day-old; $[CO_2]$ = 1200 μmol mol$^{-1}$; PPF = 750 μmol m$^{-2}$ s$^{-1}$; $T_{air}$ = 21 °C; RH = 64%) using Equation (5). In the dark, net radiation was negligible and the canopies were always cooler than air temperature because the transpiration rate and latent heat flux of hydroponic plants remains high [42]. However, the topmost leaf layers remained warm compared to the lower layers of the canopy (Figure 3A), as advection of warm air from the chamber temperature control system supplies additional energy for transpiration. Latent heat increased and sensible heat decreased in the hours preceding the photoperiod (Figure 4), probably due to circadian increases in predawn stomatal conductance [43].

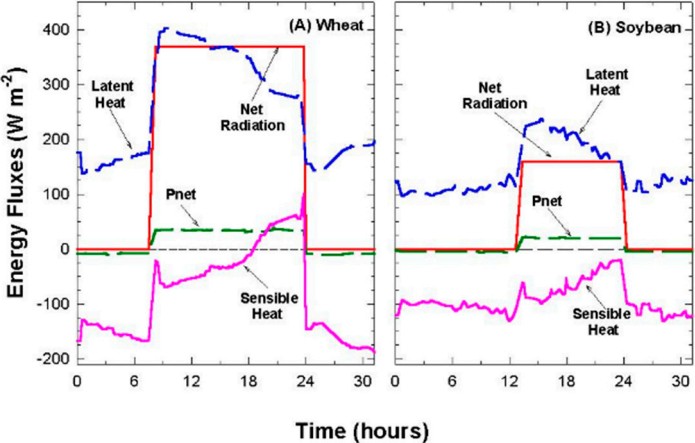

**Figure 4.** Diurnal course of canopy energy balance components: net radiation, latent heat flux, sensible heat flux, and photosynthesis in (**A**) wheat and (**B**) soybean canopies.

Generally, sensible heat increased during the photoperiod as the canopy became warmer because evaporative cooling from latent heat diminished during the course of the day, even though incident PPF was constant. This decrease in latent heat is probably due to diurnal changes in stomatal conductance [42]. In wheat, sensible heat was negative whenever latent heat plus photosynthesis exceeded net radiation, but the canopy became hotter than air temperature and sensible heat was positive at the end of the day (Figure 4A). The soybean canopy remained cooler than the air temperature, and latent heat remained greater than net radiation in spite of decreasing latent heat at the end of the day (Figure 4B).

### 3.3. Canopy-to-Air Teperature Difference

The difference between aerodynamic $\Delta T_A$ and radiometric $\Delta T_{IR}$ is affected by two physical factors: the field of view of the IR transducers and the chamber wind speed. The field of view of the sensor with respect to the canopy surface influenced the magnitude of the radiometric $T_R$ measured by the infrared transducers. Differences in $T_{IR}$ and $T_{Aero}$ are probably due to differences in how well radiometric measurements truly represent the average canopy temperature profile. With constant $T_{air}$ and $PPF_o$ provided by the chamber, radiometric $T_R$ was affected by the vertical positioning of the infrared transducers above or within the canopy. Generally, $T_R$ was higher in the surface layers of foliage and became lower as the IR transducer was inserted into the canopy foliage. Once the IR transducers were positioned, the canopy-to-air temperature difference was compared to the canopy-to-air temperature difference obtained from H.

The relation between H, $\Delta T_{IR}$, and $\Delta T_A$ was explored in soybean by changing the amount and quality of incident radiation (Figure 5; 45-day-old; [$CO_2$] = 400 µmol mol$^{-1}$; PPF = 1050 µmol m$^{-2}$ s$^{-1}$; $T_{air}$ = 22 °C; RH = 62%). In the dark, the energy balance components under each water filter were similar, yielding H ~ −75 W m$^{-2}$ (Figure 5, top), but radiometric $\Delta T_{IR}$ and aerodynamic $\Delta T_A$ differed by a nearly constant offset (Figure 5, bottom). The spikes in H observed at the beginning and at the end of the photoperiod are artifacts that occur when H is obtained by subtraction and chamber energy fluxes and temperatures equilibrate.

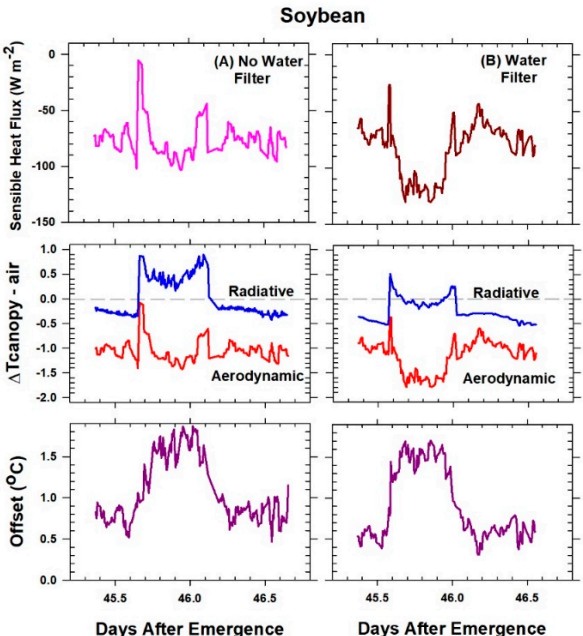

**Figure 5.** Diurnal changes in canopy sensible heat, radiative, and aerodynamic canopy-to-air temperature differences, and the offset of soybean illuminated by lamps (**A**) without and (**B**) with a water filter to remove excess longwave radiation.

During the photoperiod, removing the water filter under the HPS lamps increased $R_{net}$ by ~50 W m$^{-2}$ due to a 30% greater PPF$_o$ transmission and due to increased longwave radiation as the lamps heated the glass of the water filter. Without the water filter, the ratio of photosynthetic to non-photosynthetic shortwave radiation dropped from 83:17 to 66:34, and sensible heat flux increased up to approximately $-95$ W m$^{-2}$ (Figure 5A, top), from approximately $-120$ W m$^{-2}$ (Figure 5B, top) as the additional radiation from the lamps warmed the canopy. Radiometric $\Delta T_{IR}$ was consistently higher than aerodynamic $\Delta T_A$, and the offset was between 0.8 and 1.0 °C higher than it was in the dark. In fact, the sensible heat flux calculated from $\Delta T_{IR}$ using Equation (6) often had an opposite sign to values of sensible heat flux calculated from the energy balance equation (Figure 5, middle).

However, relative changes in the magnitude of $\Delta T_{IR}$ as a function of time paralleled the relative changes in H and $\Delta T_A$ (Figure 5, bottom), and the difference between the measured $\Delta T_{IR}$ and $\Delta T_A$ remained constant throughout the photoperiod.

Wind speed determines canopy $g_A$ and affects how the foliage warms as PPF$_o$ is increased. In a wheat canopy (25-day-old; [CO$_2$] = 1200 μmol mol$^{-1}$; $T_{air}$ = 22 °C; RH = 68%; no water filter), changes in PPF$_o$ at two chamber wind speeds were used to explore the offset between $\Delta T_A$ and $\Delta T_{IR}$ (Figure 6). At each wind speed, $\Delta T_A$ calculated from H by inverting Equation (6) was compared with values of $\Delta T_{IR}$ measured by the IR sensors.

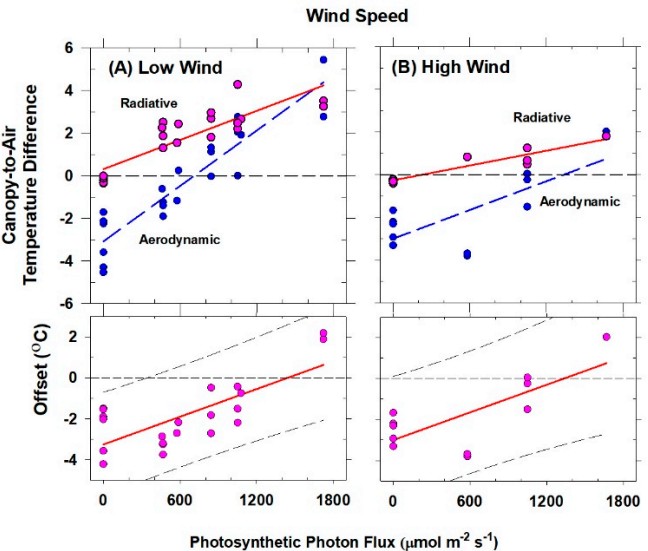

**Figure 6.** The radiometric ($\Delta T_{IR}$) and aerodynamic ($\Delta T_A$) temperatures and the offset were measured at (**A**) low and (**B**) high chamber wind speed settings.

As incident PPF$_o$ increased from 0 to 1700 μmol m$^{-2}$ s$^{-1}$, $\Delta T_{IR}$ increased linearly from 0 °C to +4 °C at low wind speed (Figure 6A, top; 1.7 m s$^{-1}$) and increased from $-1$ °C to +2 °C at high wind speed (Figure 6B, top; 2.3 m s$^{-1}$). Although the aerodynamic $\Delta T_A$ also increased linearly with increasing PPF$_o$ (Figure 6A,B), its sign was negative at low to moderate light levels and it had a steeper response to PPF$_o$ than $\Delta T_{IR}$ (e.g., changing from $-3$ °C to +4 °C at the low wind setting; Figure 6A).

The radiometric $\Delta T_{IR}$ never equaled zero when $\Delta T_A$ was zero (Figure 6) and was often opposite in sign to the aerodynamic $\Delta T_A$ (Figure 6A,B, top). The offset correction between the radiometric $\Delta T_{IR}$ and the aerodynamic $\Delta T_A$ increased linearly with increasing PPF$_o$, but did not vary with chamber wind speed (Figure 6A,B, bottom graphs; the dashed lines are the 95% confidence interval). Smaller values of the offset at high light intensities suggest that the warmer leaves at the top of the canopy play a greater role in H and reduce the differences between $T_R$ and $T_{Aero}$.

This analysis suggests that $\Delta T_{IR}$ cannot be used to determine H directly, that is, without correcting for the offset. Thus, the offset in part corrects estimates of H for differences between $\Delta T_{IR}$ and $\Delta T_A$ and allows Equations (6) and (7) to accurately describe the energy balance of dense canopies.

### 3.4. Canopy Aerodynamic Conductance

In controlled-environment chambers, $g_A$ is determined by an interaction between canopy architecture and air circulation in the chamber. Typically, fan speed is constant and canopy architecture remains constant over several days once the canopy is closed. In these conditions, Equation (6) permits canopy $g_A$ to be calculated from the slope of a plot of H versus $\Delta T_{IR}$. Stepwise increases in chamber $CO_2$ concentration from 400 to 1200 µmol mol$^{-1}$ were used to simultaneously alter H and $\Delta T_{IR}$ via physiological changes in canopy $G_S$ at a constant canopy $g_A$. H and $\Delta T_{IR}$ increase simultaneously when chamber ambient $CO_2$ increases because elevated $CO_2$ reduces stomatal conductance, and the canopy is warmed due to less evaporative cooling.

A plot of H and radiometric $\Delta T_{IR}$ was used to calculate the $g_A$ of a wheat canopy (Figure 7; 25-day-old; PPF = 1200 µmol m$^{-2}$ s$^{-1}$; $T_{air}$ = 22 °C; RH = 70%). The slopes of H versus $\Delta T_{IR}$ at each $CO_2$ concentration were similar (separate regressions not shown in Figure 7), which suggests that $g_A$ did not respond to changes in ambient $CO_2$.

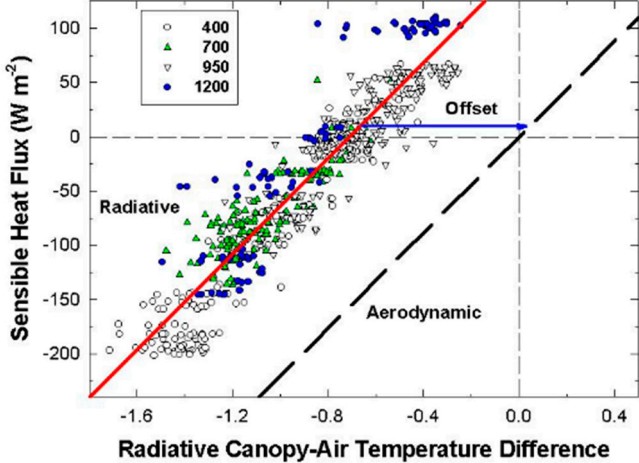

**Figure 7.** Plot of H versus $\Delta T_{IR}$ (red line) and the offset from a 25-day-old wheat canopy exposed to changing $CO_2$ concentration.

The variability in H ($\sim \pm 25$ W m$^{-2}$) corresponds to an uncertainty in $\Delta T_{IR}$ of $\sim \pm 0.4$ °C, which is close to the error in determining $T_R$ from $T_{canopy,IR}$. The dashed line in Figure 7 represents the plot of H versus $\Delta T_A$, determined by subtracting a constant offset to $\Delta T_{IR}$ (Equation (7)). This offset equals the value of the difference between $\Delta T_{IR}$ and $\Delta T_A$ when H is zero. In wheat, this offset was +0.75 °C at 1600 µmol m$^{-2}$ s$^{-1}$ and was +1.0 °C in soybean at 750 µmol m$^{-2}$ s$^{-1}$.

The $g_A$ of the 25-day-old wheat canopy was 5.5 mol m$^{-2}$ s$^{-1}$ (Figure 7). The $g_A$ of a 45-day-old soybean canopy was 2.5 mol m$^{-2}$ s$^{-1}$. These conductances correspond to aerodynamic resistances of 7.5 and 16.5 s m$^{-1}$, respectively. Soybean has a smaller $g_A$ compared to wheat because soybean leaves are wider than wheat leaves and have a smaller leaf boundary layer conductance.

### 3.5. Canopy Surface and Stomatal Conductances

Canopy surface $G_{SFC}$ of wheat was calculated from $E_{can}$ and $D_S$ using Equation (8) (green line; Figure 8A; 26-day-old; [$CO_2$] = 400 µmol mol$^{-1}$; PPF = 1600 µmol m$^{-2}$ s$^{-1}$; $T_{air}$ = 21 °C; RH = 68%). Once $g_A$ was determined, $G_{SFC}$ was used for estimating canopy $G_S$ using Equation (9) (green line; Figure 8B). Assuming that canopy $G_S$ equals $G_{SFC}$, that is, without taking $g_A$ into account (green lines in Figure 8A,B) underestimates $G_S$ by 40% in wheat.

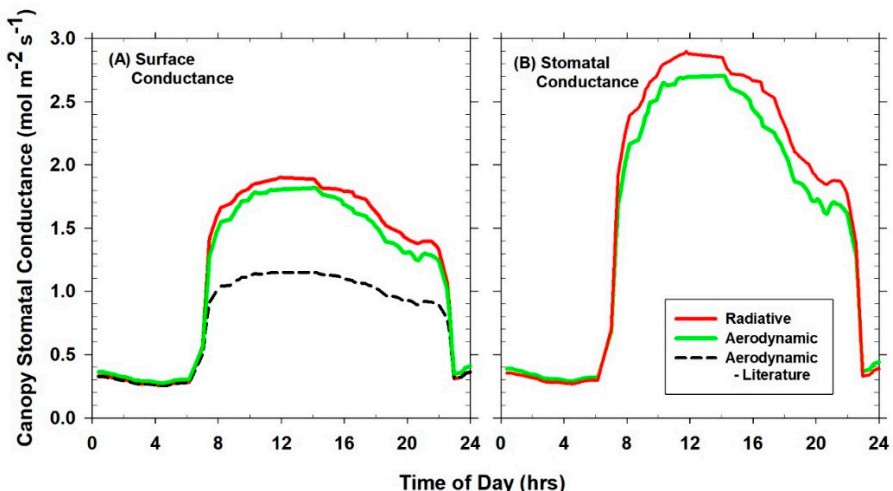

**Figure 8.** Daily courses of (**A**) canopy surface stomatal conductance ($G_{SFC}$; Equation (8)), and (**B**) canopy stomatal conductance ($G_S$; Equation (9)) of wheat.

The sensitivity of $G_{SFC}$ (Equation (8)) to errors from using $\Delta T_{IR}$ instead of $\Delta T_A$ was also explored (Figure 8). Surface $G_{SFC}$ of wheat (Figure 8A) was only slightly greater when calculated from radiometric $T_R$ instead of aerodynamic $T_{Aero}$. The average surface $G_{SFC}$ at the radiometric $T_R$ was 1.6 mol m$^{-2}$ s$^{-1}$ (red line; Figure 8A) and 1.5 mol m$^{-2}$ s$^{-1}$ (green line; Figure 8A) at the aerodynamic $T_{Aero}$. Therefore, neglecting the offset correction between $\Delta T_{IR}$ and $\Delta T_A$ in wheat resulted in only a –6% error in surface $G_{SFC}$. The difference between the average radiometric $G_S$ (2.3 mol m$^{-2}$ s$^{-1}$ or 18.0 s m$^{-1}$) and aerodynamic $G_S$ (2.1 mol m$^{-2}$ s$^{-1}$ or 19.7 s m$^{-1}$) was also small (red vs. green line; Figure 8B). Thus, canopy $G_S$ of wheat computed using the observed $T_R$ instead of $T_{Aero}$ was only 8% higher.

In soybean at 400 μmol mol$^{-1}$ of $CO_2$, the surface radiometric $G_{SFC}$ was 0.56 mol m$^{-2}$ s$^{-1}$ and aerodynamic $G_{SFC}$ was 0.75 mol m$^{-2}$ s$^{-1}$, thus using $T_R$ instead of $T_{Aero}$ to estimate surface $G_{SFC}$ of soybean resulted in a larger (−34%) error. The corresponding radiometric and aerodynamic values of canopy $G_S$ were 0.7 mol m$^{-2}$ s$^{-1}$ and 1.1 mol m$^{-2}$ s$^{-1}$, a difference of 49%.

In wheat, the sensitivity of $G_{SFC}$ to errors in $g_A$ was examined by comparing the measured $G_{SFC}$ with the $G_{SFC}$ obtained from the observed $G_S$ and a typical value of field $g_A$ reported in the literature ($g_A$ = 2 mol m$^{-2}$ s$^{-1}$; [44]). This field value of $g_A$ corresponds to less turbulent conditions (a smaller $g_A$) than were observed for wheat in the chamber, and it is closer to the $g_A$ of soybean. The $G_{SFC}$ calculated by inverting Equation (9) using measured $G_S$ and field $g_A$ (Figure 8A; dashed line) underestimates the measured $G_{SFC}$ (Figure 8A; green line) by 33%. This analysis shows that large differences in surface $G_{SFC}$ exist between field settings and controlled-environment chambers and that these occur because of differences in turbulence that can be accounted for only when $g_A$ is known.

The canopy $g_A$ and $G_S$ for wheat (18-day-old; [$CO_2$] = 400 μmol mol$^{-1}$; PPF = 1600 μmol m$^{-2}$ s$^{-1}$; $T_{air}$ = 21 °C; RH = 68%) and soybean (25-day-old; [$CO_2$] = 400 μmol mol$^{-1}$; PPF = 750 μmol m$^{-2}$ s$^{-1}$; $T_{air}$ = 21 °C; RH = 64%) canopies are reported in Table 2.

**Table 2.** Canopy $g_A$, $G_S$, and $\Omega$ from two crop architectures at 400 umol mol$^{-1}$ $CO_2$ and 86 kPa.

| Species | Architecture | $g_A$ [1] | Gs | $\Omega$ | Ds-D$_{Bulk}$ [2] | D$_{Bulk}$ |
|---------|--------------|-----------|-----|----------|-------------------|------------|
| Wheat | Erectophile | 5.5 (7.5) | 2.3 (18) | 0.67 | 0.38 | 0.74 |
| Soybean | Planophile | 2.5 (16.5) | 1.1 ( 37) | 0.39 | 0.56 | 1.15 |

[1] μmol m$^{-2}$ s$^{-1}$ (s m$^{-1}$). [2] kPa.

### 3.6. Diurnal Changes in $G_S$

Diurnal changes in $\Omega$ reflect changes in $G_S$ because chamber $g_A$ and $D_{Bulk}$ are constant (Equation (10)). The effect of $CO_2$ concentration on the diurnal course of $G_S$ was examined in wheat (Figure 9A; (18-day-old; PPF = 1600 µmol m$^{-2}$ s$^{-1}$; $T_{air}$ = 21 °C; RH = 68%)) and soybean (Figure 9B; (27-day-old; PPF = 750 µmol m$^{-2}$ s$^{-1}$; $T_{air}$ = 21 °C; RH = 64%)) canopies.

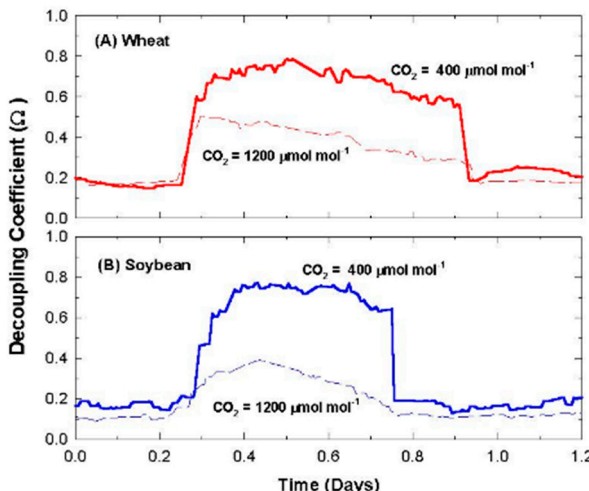

**Figure 9.** Diurnal changes in $\Omega$ of (**A**) wheat and (**B**) soybean at two chamber $CO_2$ concentrations.

In the dark, $\Omega$ of both canopies was below 0.2, Ds was coupled to $D_{Bulk}$, and canopy transpiration was small. Indeed, the nighttime VPDs within the wheat canopy ($D_{Aero}$ 1.12 kPa, and $D_S$ 1.15 kPa) were near the VPD ($D_{Bulk}$ 1.27 kPa) of the chamber. When the lights came on, the stomata opened, the chamber humidity and Gs increased, the boundary layer within the canopy became humidified by transpiration, and $D_{Bulk}$ decreased during the photoperiod. In wheat, VPDs within the canopy ($D_{Aero}$ 0.57 kPa and $D_S$ 0.36 kPa) became decoupled from $D_{Bulk}$ (0.74 kPa). At 400 µmol mol$^{-1}$ $CO_2$, the mean daily $\Omega$ of wheat was 0.67 and 0.55 in soybean (Equation (10); Figure 9) due to the differences in their corresponding $G_S$ and $g_A$. $\Omega$ during the early part of the day rose to ~0.8 in wheat and to ~0.75 in soybean. As the photoperiod progressed, $\Omega$ gradually declined to ~0.6 in wheat and to ~0.6 in soybean as a consequence of a diurnal decrease in $G_S$. At 1200 µmol mol$^{-1}$ $CO_2$, $\Omega$ of wheat and soybean reached a maximum of 0.4 due to reduced stomatal conductance and declined to near 0.2 at the end of the photoperiod (Figure 9).

### 3.7. Control of Canopy Transpiration by $CO_2$ Concentration

The effect of $CO_2$ concentration on canopy transpiration and $\Omega$ was explored using a wheat canopy (27 to 34 day old; PPF = 1600 µmol m$^{-2}$ s$^{-1}$; $T_{air}$ = 21 °C; RH = 68%) exposed to varying chamber $CO_2$ concentrations ranging between 400 and 1200 µmol mol$^{-1}$ (Figure 10; Table 3).

The $CO_2$ concentration was raised in steps from 400, to 700, to 950, and to 1200 µmol mol$^{-1}$ and allowing a 48 h acclimation period at each $CO_2$ concentration. Simulated decoupling coefficients were calculated for increasing values of $g_A$ (Figure 10; 2 mol m$^{-2}$ s$^{-1}$ (dotted line), 4 mol m$^{-2}$ s$^{-1}$ (dashed line), and 8 mol m$^{-2}$ s$^{-1}$ (solid line)). The decoupling coefficient increased as canopy $G_S$ increased, reaching an average $\Omega$ of 0.65 with a canopy $G_S$ of 2.1 mol m$^{-2}$ s$^{-1}$ at 400 µmol mol$^{-1}$ $CO_2$ (Figure 10; Table 3). Increasing $CO_2$ concentration from 390 to 690 µmol mol$^{-1}$ led to 1.77 X $CO_2$ or nearly a doubling in ambient $CO_2$. This change decreased mean daily $G_S$ by −35%, resulting in a −23% lower transpiration rate and a -23% reduction in latent heat (Table 3). Since $g_A$ remained constant, this decrease in $G_S$ caused a −26% decrease in $\Omega$. These results indicate that $E_{can}$ is less sensitive to changes in stomatal conductance due to the decoupling of $D_s$ from $D_{Bulk}$. In addition, an increase in $P_{net}$ of 15% and a −23% decrease in $E_{can}$ resulted in a 150% increase in WUE (Table 3).

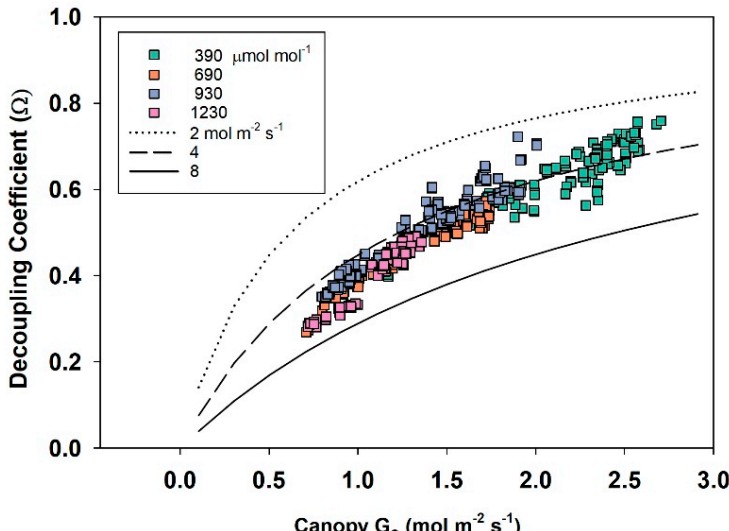

**Figure 10.** The decoupling coefficient, $\Omega$, expressed as a function of canopy $G_S$ in wheat measured at $CO_2$ concentrations ranging between 400 and 1200 $\mu$mol mol$^{-1}$.

**Table 3.** Canopy level responses to elevated $CO_2$ at constant PPFo and $g_A$.

| Parameter | Symbol | Units | $CO_2$ Concentration ($\mu$mol mol$^{-1}$) | | | |
|---|---|---|---|---|---|---|
| | | | 390 | 690 | 930 | 1230 |
| Transpiration | $E_{can}$ | mmol m$^{-2}$ s$^{-1}$ | 11.2 | 8.7 | 7.9 | 7.0 |
| | | % | 100 | 77 | 71 | 63 |
| Photosynthesis | $A_{can}$ | $\mu$mol m$^{-2}$ s$^{-1}$ | 63 | 73 | 81 | 79 |
| | | % | 100 | 115 | 128 | 125 |
| Latent Heat | LE | W m$^{-2}$ | 460 | 356 | 324 | 288 |
| Sensible Heat | H | W m$^{-2}$ | $-175$ | $-70$ | $-42$ | $-3$ |
| Stomatal Conductance | $G_S$ | mmol m$^{-2}$ s$^{-1}$ | 2145 | 1394 | 1152 | 1031 |
| | | % | 100 | 65 | 54 | 48 |
| Decoupling coefficient | $\Omega$ | dim. | 0.65 | 0.48 | 0.45 | 0.39 |
| | | % | 100 | 74 | 69 | 60 |
| Water Use Efficiency | WUE | $\mu$mol mmol$^{-1}$ | 5.6 | 8.4 | 10.2 | 11.2 |
| | | % | 100 | 150 | 183 | 199 |
| Changes in $E_{can}$, Gs, LE as $CO_2$ increased from 400 $\mu$mol mol$^{-1}$ | dEcan | mmol m$^{-2}$ s$^{-1}$ | - | 2.5 | 3.3 | 4.2 |
| | dLE | W m$^{-2}$ | - | 104 | 136 | 172 |
| | $dG_S$ | mol m$^{-2}$ s$^{-1}$ | - | 0.75 | 0.99 | 1.11 |
| | $dE_{can}/dG_S$ | mmol mol$^{-1}$ | - | 3.33 | 3.32 | 3.75 |

The relative changes in $G_S$, $E_{can}$, and $\Omega$ of wheat as $CO_2$ concentration increased, from Table 3, are shown in Figure 11. Canopy $G_S$ decreased by 52%, but $E_{can}$ only decreased by 37% when $CO_2$ concentration was raised from 390 to 1230 $\mu$mol mol$^{-1}$ because of the feedback between $E_{can}$ and $D_S$. In these chamber settings, $E_{can}$ is much less sensitive to a proportional change in $G_S$ and the reduction in $E_{can}$ is largely explained by $\Omega$.

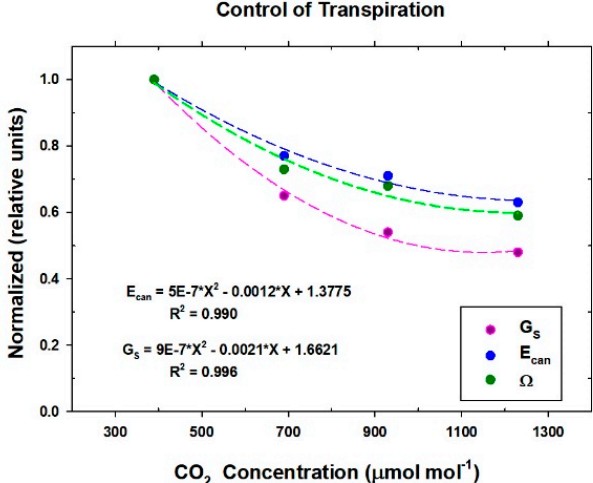

**Figure 11.** Relative changes in $G_S$, $E_{can}$, and $\Omega$ as chamber $CO_2$ concentration increases.

## 4. Discussion

### 4.1. Canopy Stomatal Conductance

Land components of climate and carbon models require accurate descriptions of the stomatal control of canopy energy exchange, evapotranspiration, and carbon exchange because land surfaces provide a continuous feedback of latent and sensible heat fluxes to the atmosphere, which drives weather and climate [45]. The method developed in this study expands the usefulness of controlled environments for improving land surface models because it allows the measurement of responses of canopy-level $G_S$, an essential control of canopy gas exchange, to environmental variables.

In this study, $G_S$ was measured when surface radiation forcing ($R_{net}$), boundary layer forcing ($T_{air}$ & $D_{Bulk}$), surface layer feedbacks ($g_A$), and soil moisture were held constant during the photoperiod. Furthermore, the use of well-watered plant stands grown at constant light reduced much of the environmental variability that confounds estimates of $G_S$ in natural ecosystems, such as periodic drought, the diurnal change in solar radiation, or short, temporal fluctuations in radiation due to cloud cover. Moreover, the carbon and water vapor fluxes measured in this study do not include significant contributions from soil respiration and evaporation as compared to field measurements.

Canopy $G_S$ was derived from direct measurements of surface $G_{SFC}$, $g_A$, and energy balance ($R_{net}$, LE, and P) in controlled environments. Once the IR transducers were positioned above the canopy, chamber $CO_2$ concentration was manipulated to alter stomatal conductance, which in turn resulted in corresponding changes in sensible heat flux and the canopy–air temperature difference. Canopy $g_A$ was obtained radiometrically from the slope of a plot of H vs. $\Delta T_{IR}$ (Figure 7), and the offset correcting for differences between $T_R$ and $T_{aero}$ was determined.

The radiometric method presented here differs from other methods for calculating $G_S$ because canopy $g_A$, $E_{can}$, and canopy-to-air temperature differences are measured directly. This avoids the complexity of methods for scaling leaf level observations to the canopy scale because these must integrate responses of leaf stomatal conductance to vertical profiles in radiation, temperature, humidity, and wind speed within a canopy. Separating $g_A$ from the measured canopy $G_{SFC}$ permits the determination of the physiologically controlled canopy-scale $G_S$, which is equivalent to the "big-leaf" stomatal conductance, where the stomatal conductances of individual leaves of the canopy act in parallel, and the vertical gradients in temperature and humidity are averaged by the aerodynamic $T_{Aero}$ and $D_{Aero}$. The strength of this approach is that canopy $G_S$ responses are measured at the correct scale for predicting ET in future elevated $CO_2$ and climate change scenarios. Furthermore, estimates of canopy ET made using the measured $G_S$ also account for the feedback of Ds on ET, as shown by the decoupling coefficient.

In the chamber, $g_A$ was set constant by the air flow rate provided by its recirculation fans. However, the $g_A$ established in the chamber was different for each species when measured at the same turbulent field provided by the chamber fans. The $g_A$ for the wheat (5.5 mol m$^{-2}$ s$^{-1}$ or 7.5 s m$^{-1}$) and for the soybean (2.5 mol m$^{-2}$ s$^{-1}$ or 16.5 s m$^{-1}$) canopies were within the range of typical aerodynamic conductances of field crops (ranging from 3.2–10 mol m$^{-2}$ s$^{-1}$ or 4–13 s m$^{-1}$; [46]). A smaller $g_A$ for soybean compared to wheat reflects a larger canopy boundary layer associated with the broader soybean leaves.

In this study, $G_S$ values of wheat and soybean were measured at an elevation of 1460 m (4800 ft), a barometric pressure of 86 kPa, and 400 µmol mol$^{-1}$ $CO_2$ (Table 2). For wheat, $G_S$ was 2.3 mol m$^{-2}$ s$^{-1}$ or 18 s m$^{-1}$, which is slightly higher than field $G_S$ values (1.8 mol m$^{-2}$ s$^{-1}$ or 22.7 s m$^{-1}$) reported by Hatfield [47] at sea level, under optimal available soil water. The $G_S$ of soybean was also slightly higher than typical conductances measured in field crops [46]. Soybean $G_S$ at 400 µmol mol$^{-1}$ $CO_2$ was 1.1 mol m$^{-2}$ s$^{-1}$ or 37 s m$^{-1}$, nearly one-half the value found in wheat probably due to less leaf area and because it was measured at a lower $PPF_o$. The $G_S$ values of this study are expected to be higher than those measured at sea level because, at lower atmospheric pressures, the diffusion coefficients of water vapor and $CO_2$ in air increase, so $G_S$ also increases [48,49].

*4.2. The Control of Transpiration by CO$_2$ Concentration*

The radiometric method developed in this study was used to determine the response curve of $G_S$ to $CO_2$ concentration in wheat (Figure 11, Table 3). As $CO_2$ concentration increased, the measured decrease in $E_{can}$ was lower than the measured decrease in $G_S$ because feedback between $E_{can}$ and $D_S$ operating at the canopy scale effectively reduces the sensitivity of $E_{can}$ to changes in $G_S$. Thus, a smaller change in $E_{can}$ was observed as $CO_2$ increased, and the reduction in $E_{can}$ is largely explained by changes in $\Omega$, which are determined by the relative magnitudes of $g_A$ and $G_S$.

In this study, an increase of 1.77 X $CO_2$ (that is, an increase from 390 to 690 µmol mol$^{-1}$; Table 3) caused a −23% drop in $E_{can}$ and a +15% increase in photosynthesis. These changes are comparable to the results of Friend and Cox [50], who used a combined climate-vegetation model to predict a similar −25% drop in ET and a +19.4% increase in GPP for a doubling ambient $CO_2$ (2 X $CO_2$). In a four-year SoyFACE study, Bernacchi et al. [51] found a 9–16% reduction in canopy ET and reported that meta-analyses across FACE experiments indicate a 17–22% drop in leaf level stomatal conductance when daytime $CO_2$ was raised by 175 umol mol$^{-1}$ from 375 to 550 umol mol$^{-1}$. In this study, the two regression equations in Figure 11 ($E_{can}$ and $G_S$ as a function of $CO_2$ concentration) predict a 13% decrease in $E_{can}$ and a 22% decrease in canopy $G_S$ for an increase of 175 umol mol$^{-1}$ of $CO_2$. These comparisons suggest that the responses of $G_S$ and $E_{can}$ to $CO_2$ reported in this study are similar to responses in canopy $G_S$ and ET observed in $CO_2$-enriched plant canopies in field settings.

## 5. Conclusions

The controlled-environment experiments conducted in this study provide a new methodology for measuring canopy stomatal and aerodynamic conductances. A radiometric method for determining canopy aerodynamic conductance from changes in vegetation temperature and energy balance was developed in controlled environments using a canopy-level gas exchange system. The gas exchange system measured canopy gas fluxes ($CO_2$ and water vapor), energy balance (net radiation, latent and sensible heat fluxes), and canopy temperatures as $CO_2$ concentration was varied. Two key assumptions of this method are that radiative canopy temperature is approximated by canopy brightness temperature and that the difference between aerodynamic and radiative canopy-to-air temperature differences is constant during the photoperiod. Once canopy aerodynamic conductance was determined from a plot of sensible heat flux versus the radiative canopy-to-air temperature difference, canopy stomatal conductance was calculated from measurements of canopy transpiration. The method was used to determine the curves of the response of canopy stomatal conductance and canopy ET to increased $CO_2$ concentration in wheat (Table 3; Figure 11). Predictions of canopy ET

made from the measured response of $G_S$ to elevated $CO_2$ are comparable to land surface model predictions and to observed changes in ET found in FACE studies [50,51]. Future work should focus on studying how canopy stomatal conductance measured using this methodology responds to drought, vapor pressure deficit, and temperature to provide data sets for calibrating global climate models. The method should also be used to characterize how canopy aerodynamic conductance changes during the growth cycle of different crop species.

**Author Contributions:** O.M. and B.B. conceived the study. O.M. wrote the original manuscript. O.M. and B.B. reviewed and revised the manuscript. Project administration and funding acquisition: B.B. All authors read and approved the manuscript.

**Funding:** This research was supported by the Advanced Life Support Program of the National Aeronautics and Space Administration and by the Utah State Agricultural Experiment Station, Utah State University. This manuscript has been approved as journal paper number 9178.

**Acknowledgments:** The authors would like to thank John Norman, Marc Van Iersel, and Larry Hipps for critically reviewing the manuscript.

**Conflicts of Interest:** The authors declare no conflict of interest. The funders had no role in the design of the study; in the collection, analyses, or interpretation of data; in the writing of the manuscript; or in the decision to publish the results.

## Abbreviations

| Symbol | Description | Units |
|---|---|---|
| $C_p$ | Heat capacity of air at constant pressure | kJ m$^{-3}$ °C$^{-1}$ |
| [$CO_2$] | $CO_2$ concentration | umol mol$^{-1}$ |
| DAE | Days after emergence | d |
| E | Chamber evaporation rate | mmol m$^{-2}$ s$^{-1}$ |
| Ecan | Canopy transpiration rate | mmol m$^{-2}$ s$^{-1}$ |
| ET = $\Delta X_{h20}$*MF | Chamber evapotranspiration | mmol m$^{-2}$ s$^{-1}$ |
| $g_S$ | Single leaf stomatal conductance | mol m$^{-2}$ s$^{-1}$ |
| $g_A$ | Canopy aerodynamic conductance | mol m$^{-2}$ s$^{-1}$ |
| G | Soil heat flux | W m$^{-2}$ |
| $G_{SFC}$ | Canopy surface conductance | mol m$^{-2}$ s$^{-1}$ |
| $G_S$ | Canopy stomatal conductance | mol m$^{-2}$ s$^{-1}$ |
| H | Sensible heat flux | W m$^{-2}$ |
| LE | Latent heat flux | W m$^{-2}$ |
| ↑Lc | Longwave radiation emitted by the canopy | W m$^{-2}$ |
| ↓Lg | Longwave radiation emitted by the glass of the water filter | W m$^{-2}$ |
| MF | Mass flow rate of air | mol s$^{-1}$ |
| $NPSW_{abs}$ | Absorbed non-photosynthetic shortwave radiation | W m$^{-2}$ |
| Offset | Difference between $\Delta T_A$ and $\Delta T_{IR}$ | °C |
| P | Energy storage in photosynthesis | W m$^{-2}$ |
| $P_{net}$ | Canopy net photosynthetic rate | µmol m$^{-2}$ s$^{-1}$ |
| $PPF_o$ | Incident photosynthetic photon flux | µmol m$^{-2}$ s$^{-1}$ |
| $PPF_{abs}$ | Fraction of incident PPF absorbed by the canopy | µmol m$^{-2}$ s$^{-1}$ |
| $R_{net}$ | Net radiation | W m$^{-2}$ |
| s | Slope of the relation between saturation vapor pressure and temperature | |
| $T_{air}$ | Mean air temperature measured above the canopy | °C |
| $T_{Aero}$ | Canopy aerodynamic temperature | °C |
| $T_{canopy,IR}$ | Canopy brightness temperature measured by IR transducers | °C |
| $T_{glass}$, $T_{wall}$ | Water filter glass and chamber wall temperatures | °C |
| $T_R$ | Canopy radiometric temperature | °C |
| $T_{Sky}$ | Composed of 20% chamber $T_{wall}$ and 80% $T_{glass}$ | °C |

| $\gamma$ | Psychrometric constant | $kPa\ K^{-1}$ |
|---|---|---|
| $\Delta T_A$ | Aerodynamic canopy-to-air temperature difference ($T_{Aero} - T_{air}$) | $^\circ C$ |
| $\Delta T_{IR}$ | Radiometric canopy-to-air temperature difference ($T_R - T_{air}$) | $^\circ C$ |
| $\Delta X_{h20}$ | Mole fraction difference between pre- and post-chamber water vapor | |
| $\varepsilon = s/\gamma$ | ratio of the increase of latent heat content to the increase of sensible heat content of saturated air | |
| $\varepsilon_c$ | Canopy emissivity | |
| $\varrho$ | Density of air | $kg\ m^{-3}$ |
| $\varrho_c$ | Canopy reflection coefficient | |
| $\sigma$ | Stefan–Boltzman constant | $W\ m^{-2}\ K^{-4}$ |
| $\sigma_S$ | Scattering coefficient | |
| $\Omega$ | Decoupling coefficient | |
| $X_{H2O}(T_{air})$ | Mol fraction of water vapor at $T_{air}$ above the canopy | |
| $X_{H2O}(T_{Aero})$ | Mol fraction of water vapor at $T_{Aero}$ | |

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
