# Peer review of "Radiometric Method for Determining Canopy Stomatal Conductance in Controlled Environments"

_agronomy, doi:10.3390/agronomy9030114_

Round 1
Reviewer 1 Report
General comments
I enjoyed reading the manuscript. The subject under consideration is interesting and useful to the readers of Agronomy. This manuscript has brought a new methodology of estimating the canopy stomatal conductance for two agriculture crop species: wheat and soybean under the controlled environment chambers, where net radiation, temperature, humidity, and ventilation could be kept under certain limit as needed. The proposed method is useful to determine the response of canopy stomatal conductance to the increased CO2 concentration and determine the sensitivity of canopy transpiration to changes in canopy stomatal conductance. These responses are expected to be useful for improving the prediction of ecosystem-level water fluxes in response to climatic variables. The study is relevant and timely in the context of climate change. The manuscript is written well, except few problems of using inconsistent terminologies, and over-use symbols and abbreviations, which are not necessary in many places. The proposed method is naïve, and therefore I consider this manuscript has a good merit to be published in the Agronomy. However, authors need to revise the manuscript by considering the following issues, before it will go for further evaluation.
Minor issues
1) Authors are suggested to avoid-or if not possible, minimize the usage of symbols and abbreviations in the manuscript, especially in abstract and conclusion section, they should not stay at all as these sections are considered independent of other sections of the manuscript for most of the readers, who are quite busy and want to understand your messages without troubling of searching the meaning of symbols and abbreviations in other sections of the manuscript. For example, symbol Gs for canopy stomatal conductance, GSFC for bulk surface conductance to water vapor, GA for canopy aerodynamic conductance, and so on are not necessary in the abstract and conclusions. Also, unless deemed necessary, please minimize the uses of the symbols and abbreviations in other parts of the manuscript as well. Also, please make consistent use of the words throughout the manuscript.
2) Line 12: ‘plant canopy’
3) Line 17: please don’t start sentence with abbreviation, symbol and number, you may add ‘The’ before H here and other places as well. However, here H is not necessary, please replace it with sensible heat.
4) Lines 12-15: please mention wheat and soybean in the objective. It may be like this-“The objective of this study was to develop a radiometric method for calculating canopy stomatal conductance for two plant species- wheat and soybean from direct measurements of bulk surface conductance to water vapor and the canopy aerodynamic conductance in the controlled environment chambers”
5) Line 36: Please replace Gs with its definition, and please consider avoiding or minimizing similar symbols and abbreviations from introduction section as well. These can be used for method section.
6) Line 94: plant canopy
7) Line 98: please make italics for two species names.
8) Line 93-101: Please state that what hypothesis did you assume for this study, and how and why your proposed method is more important than other existing ones for determining Gs?
9) Line 108: delete ‘germinated and’
10) Method section is too long and readers may want to look at this in any of the summary form, such as theoretical frame work of method (may be graphical). Could you please present such as frame work in in the beginning or end of the method section?
11) Line 363, 453, please define symbols and abbreviations in Table and figure captions here and other places as well.
12) Line 379: Is it necessary to use different symbols (dots, circle) in this figure. You may use black dots for both.
13) Line 453: (a) without water filter. If (a) and (b) are defined in captions, it is not necessary to keep their verbal definitions inside the figure.
14) Line 576: why parenthesis here? And also check this in Table 3.
15) Line 577: next to “parameter” column head, “symbol and abbreviation” column head is needed.
16) Line 591: In this figure inside, there is equation, in which Gs describes 100% variations of x. How is it possible? I don’t believe this, as no independent variables of the interest in the modelled phenomena is able to describe 100% variations of dependent variable (R2=1). Please consider checking your analysis.
17) Please consider adding few sentences to discussion section that how your study is different from other methods of calculating Gs, and why it is important in elevated CO2 concentration and climate change context.
18) As said earlier, please avoid all symbols and abbreviations in the conclusion section, and this section independent section of the manuscript for busy readers to get your message very quickly.
Author Response
see attached pdf

Reviewer 2 Report
This manuscript presents a method for estimating canopy stomatal conductance in a controlled laboratory setting. The method appears to be novel and is clearly the result of a substantial amount of thought and laboratory effort. After reading the manuscript twice, I cannot find any serious errors in logic or scientific approach. In my opinion, the manuscript is exceptionally clear, interesting, and well-written. I recommend that the manuscript be accepted for publication pending minor revisions to be made at the authors’ discretion.
General Suggestions:
1. Given the number of symbols used throughout the text, the reader may benefit from distillation of their meanings into a single table.
2. It would be useful for the reader to visualize the laboratory setup if a figure were added showing photos.
3. The labels in some of the figures are unclear. These figures would benefit from use of color and/or migration of some arrows/text labels. Specific instances are given below.
4. Readers may be interested in the addition of further content in the discussion on the topics of a) potential relevance for field applications, particularly in the context of remote sensing; b) evolution of gA as the canopy evolves over the course of the plant life cycle; c) expected generality of the radiometric vs aerodynamic temperature offset for other crops and in other environments.
Line-specific comments:
Lines 104-124. It would be useful to supplement this description with photos of the laboratory setup. These could be placed in an appendix if the authors feel it is necessary to maintain the flow of the text. In particular, if available, top-down photos of the samples at varying growth stages would help the reader visualize the plant spacing & canopy closure.
Lines 120-122. The statement “The water filter under the lamps was removed to increase the incident PPFo and longwave radiation on the canopy and alter surface radiation forcing” seems to imply that the water filter was removed throughout the experiment. Given the rest of the manuscript, I assume this is not true. It might be helpful to add a qualifier explaining that the water filter was only removed on particular occasions.
Line 125. Is an estimate of the uncertainty of the barometric pressure measurement available?
Lines 134-137. How frequently was chamber CO2 cycled? Was this done on a continuous 8 day loop throughout the experiment, or only at prescribed intervals?
Lines 164-171. What was the vertical spacing of the wind speed measurements?
Lines 204-210. This is a particularly interesting point.
Line 256. Is the 45 C temperature of the glass a brightness temperature?
Line 262-263. How was it determined that evaporation from the hydroponic tubs was ~2% of Rnet?
Lines 288-303. I had to read this section several times to understand it. It may help readers if it were reworded.
Line 338. It may be useful to add a sentence here explaining the basic principle behind the resistance subtraction method so the reader can understand the concept without pulling up the reference.
Lines 349 – 353: It may be useful to add a sentence here explaining the origin & relevance of epsilon.
Lines 354 – 364: Ωc should be defined. How does Ωc differ from Ω? From equation 10, it would seem that a high value of gA would imply a low value of Ω, given that gA is in the denominator. The table shows the opposite. Is Ωc (1-Ω)?
Lines 375-376: Is the average wind speed the mean or median? The median may make more sense given the non-Gaussian nature of wind speed distributions.
Figure 1. What are the units of normalized height? Is it meters above/below the top of canopy? Fraction of canopy height?
Figure 2. This is a very interesting plot, but it is difficult to read because of the way it is labeled. The labels need to be moved and/or colorized. If each curve were made its own color (or shade of gray), and the text label color were changed to match the curve color, this would eliminate the need for the arrows and clarify which text label corresponds to which curve. It could also help to make the filled curves partially transparent.
Lines 400 – 415. Is there a reason why 20-day old plants were used for the light intensity demonstration, but 35-day old plants were used for the fan speed demonstration?
Lines 417 – 434, and Figure 3: This is another very interesting plot. It would be interesting to hear further discussion about the generality (or lack thereof) of the diurnal variability in H vs LE in either this or a subsequent work. It would also be interesting if the authors could speculate about the increase in LE (decrease in H) that occurs in the hours immediately preceding the photoperiod. Also, it might help the reader if the terms used in the text vs in the figure were standardized. For instance, either the magenta curve could be labeled “H” instead of “sensible heat”, or the text could be revised to say “sensible heat” instead of “H”. Also, it should again be explained why these particular plant ages were chosen. Is it just that the data from these days is easiest to interpret, or is there another reason?
Lines 440 – 444. How much did TR change w/ vertical positioning of the IR transducers?
Figure 4. It might be interesting to plot a histogram and/or time series of offsets to give more information about the extent to which it varies. This would allow you to quantify the statement in lines 462-464 that the offset remained constant throughout the day. Also, same point as before about the choice of 45-day old plants, and about consistency of symbols vs full variable names.
Figure 5. For the bottom row, it might be useful to add the mean +/- SD error bars for each set of points at a given PPF. In addition to the
Figure 6. This is a convincing plot.
Figure 7. This plot is also convincing. However, the description of the curves in the text as dashed vs thin vs thick lines does not match the appearance of the plot. This should be fixed.
Lines 595-639. Is there anything that we can generalize from
the results of this study to assist with ET estimation in field settings?
Author Response
see attached pdf
